# Airbrushed PSF/ZnO Composite Coatings as a Novel Approach for the Consolidation of Historical Bones

**DOI:** 10.3390/nano13040625

**Published:** 2023-02-04

**Authors:** Monireh Moradienayat, Javier González-Benito, Dania Olmos

**Affiliations:** Department of Materials Science and Engineering and Chemical Engineering, Instituto de Químicay Materiales Álvaro Alonso Barba (IQMAA), Universidad Carlos III de Madrid, 28911 Leganés, Spain

**Keywords:** polysulfone, zinc oxide (ZnO), airbrushing, mechanical properties, historical bone

## Abstract

In this work, the preparation and characterization of films based on polysulfone (PSF) filled with zinc oxide, ZnO, nanoparticles (NPs) are conducted. The novelty of this research mainly relies on two points: (i) the use of a commercial airbrush to prepare or modify materials, and (ii) the design of new materials (nanocomposites) for the consolidation and restoration of historical bones. To accomplish these objectives, free-standing thin films and ancient bone coatings of PSF/ZnO nanocomposites with different particle contents (0%, 1%, 2%, 5% and 10%, % wt) are prepared using a commercial airbrush. Mechanical characterization is carried out to correlate properties between free-standing thin films and coatings, thus understanding the final performance of the coatings as consolidants for ancient bones. Thin films of PSF/ZnO show that the elastic modulus (E) increases with particle content. The mechanical behavior of the surfaces of the treated and untreated bones is studied locally using Martens hardness measurements. Maximum values of Martens hardness are obtained for the bone samples treated with polysulfone filled with 1% ZnO nanoparticles (HM = 850 N·mm^−2^) or 2% ZnO (HM = 625 N·mm^−2^) compared to those treated just with neat PSF (HM = 282 N·mm^−2^) or untreated bone (HM = 140 N·mm^−2^), indicating there is a correspondence between rigidity of free-standing films and hardness of the corresponding coatings. In terms of mechanical performance, it is demonstrated the existence of a balance between nanoparticle concentration and probability of nanoparticle aggregation, which allows better material design for ancient bones consolidation.

## 1. Introduction

The search for different approaches for the conservation of artworks and buildings, such as paintings, sculptures, decorating objects, furniture, bones and other objects that have lost their usefulness or mechanical features due to human usage or aging process, is referred to as artwork conservation and restoration [1,2,3,4]. Because of deterioration, certain natural materials need to be fixed or even replaced. However, in many situations, the use of original natural material is not economically profitable due to its low abundance in nature or the difficulties of extracting or processing it. Therefore, consolidants have been extensively used for the consolidation of degraded artefacts [5]. In the case of bones, different polymers have been used for their consolidation, for instance, poly(vinyl acetal), poly(vinyl butyral) and Paraloid B72 [6,7]. Yet, some previous results with these polymers have shown that they have poor penetration and resistance; besides, the bone might become brittle after a long time [6,7,8].

One interesting polymer to be considered for bone restoration and consolidation is polysulfone. Polysulfone is a thermoplastic polymer with remarkable properties for this potential application, such as high thermal and chemical stability and high mechanical strength [9,10]. In addition, polysulfone is biocompatible and has been tested in animal models for bio-related applications such as bone substitute material in implants [11], suggesting that PSF is an interesting polymer to be considered for bone consolidation and restoration applications.

Modification of polymers by introducing inorganic nanoparticles is one common strategy to modify their properties and performance by changing, for instance, polymer structure, morphology and even with the appearance of synergistic effects because of the simple presence of a new phase with highly different properties. One of the main consequences of the presence of a harder filler is an increase in mechanical properties. Among the inorganic compounds used in the form of nanoparticles to fill polymers, ZnO has received much attention because of its high chemical stability, low cost, excellent antimicrobial, antifungal and UV filtering properties. Furthermore, it has a high thermal and mechanical stability [12,13]. Zinc oxide nanoparticles have been extensively used as prosthetic devices [14] and hard tissue replacements for dental composites [15]. Moreover, other uses and applications of ZnO that can be found in literature include other applications, such as restoration and conservation of paper [16], protection of stone material [17], historical building [18], leather [19] and wood coating for outdoor applications [20].

Polysulfone by itself cannot satisfy all the requirements to be applied in bone restoration and consolidation, such as mechanical performance as well as antimicrobial and UV filtering properties. In previous studies, by utilizing ZnO as filler for PSF polymer matrix, several beneficial characteristics were maintained, such as ease of processing, good moldability, fouling resistance for water treatment or even antibacterial properties under certain conditions [21,22,23,24,25,26].

Nanocomposites have shown several advantages. One of the most well-known is the fact that the use of small-scale fillers, like nanoparticles, contributes to the creation of larger interphases that can considerably modify the properties of the material, leading to the formation of large interphases that can considerably modify the properties of the polymer matrix. Fillers played a significant role in the polymer matrix, which supplies new properties and thus better performance among the existing materials. Considering the potential application of these materials for historical bone repair, a system based on polysulfone modified with the multifunctional features of the photocatalytic activity of zinc oxide nanoparticles was chosen to fulfill this major purpose.

Several techniques of consolidation, such as brushing and immersion in solutions, have been utilized for the restoration of historical artefacts in the past [5,27,28]. The advantage of immersion is that usually higher penetration depths are achieved; however, this treatment is in general only convenient for small specimens. On the other hand, brushing is an easy method to apply consolidants on the artefacts surfaces although it does not lead to enough penetration. For this reason, methods for gathering the most important advantages of those conventional processes for consolidation should be investigated. Among the methods used to coat large surfaces, those based on spraying suspensions or solutions could be very interesting in the field of restoration and/or consolidation of historical artefacts. As an example, airbrushing could be highlighted because it is a versatile method that can be easily used to coat and treat degraded surfaces in-situ. Airbrushing involves spraying the polymer solution or suspension on the desired surface, controlling the amount and consequently the thickness of the material deposited. Moreover, airbrushing and solution spraying methodologies have been revealed as good methods to obtain polymer-based nanocomposites with uniform nanoparticle dispersions [29,30,31,32,33,34,35], which is essential to prepare homogeneous materials.

The aim of this work is to prepare by airbrushing free-standing films of polymer composites based on polysulfone (PSF) filled with zinc oxide (ZnO) nanoparticles as model systems or reference materials. Then, they will be characterized for correlating the information obtained with the properties associated with bones treated with the same materials by their deposition using a commercial airbrush. To understand the effect of the presence and concentration of nanoparticles on materials’ performance, a deep study in terms of morphology (nanoparticle dispersion) and mechanical and thermal behaviors is carried out.

## 2. Experimental Part

### 2.1. Materials

Polysulfone (PSF) was purchased from Sigma-Aldrich (average number and weight molar masses, M_n_ ~ 16,000 and M_w_ ~ 35,000 g·mol^−1^ respectively, density 1.24 g·mL^−1^ at 25 °C) (Sigma-Aldrich, St. Luis, MO, USA). Commercial zinc oxide (ZnO) nanopowder was used as the filler, also from Sigma-Aldrich, with the following details: product number: 544906; CAS number: 1314-13-2; MDL: MFCD00011300; batch number: MKCG5504 and mean diameter ≤ 100 nm, Pcode: 1003034414 (Sigma-Aldrich, St. Luis, MO, USA). The certificate of analysis (COA) of this batch provided an average particle size of 67 nm. Tetrahydrofuran (THF) of HPLC quality was used as the solvent to prepare the solutions to be airbrushed.

X-ray diffraction (XRD, Phillips X’Pert diffractometer from Malvern Panalytical Ltd, Malvern, UK) and field emission scanning electron microscopy (FESEM, TENEO from FEI) were used for the structural and morphological characterization, respectively, of the as-received ZnO nanoparticles. Experimental details on X-ray diffraction and SEM experiments are provided in Section 2.3. In Figure 1, the XRD pattern (Figure 1a) and a scanning electron microscopy micrograph of ZnO powder (Figure 1b) are shown. The diffraction pattern of the commercial ZnO nanopowder (Figure 1a) was similar to that of the Joint Committee on Powder Diffraction Standard for ZnO (file no: 043-0002); labels shown over the peaks represent the planes causing the corresponding reflections. The SEM micrographs of the ZnO nanoparticles (see Figure 1b) allowed us to confirm particle size as specified by the supplier (less than 100 nm). In Figure 1c, the particle size distribution obtained from SEM micrographs is shown. The average particle size measured from SEM images, 52 nm, was in accordance with the estimated crystallite size using the Scherrer equation, approximately 42 nm.

### 2.2. Sample Preparation

All the materials were prepared from PSF solutions in THF with a concentration of 5% (%, in g·mL^−1^). For the preparation of PSF/ZnO nanocomposites, suspensions of ZnO nanoparticles were mixed with PSF solutions according to the following protocol. First, the solutions were obtained by dissolving 0.5 g of PSF in 10 mL of THF (5% g·mL^−1^). The nanocomposites were prepared from a suspension made by mixing a solution of 0.5 g of PSF in 7 mL of THF with 3 mL of THF (to keep constant the concentration of the PSF solution) containing the required suspended amount of ZnO nanoparticles so as to finally have a nanocomposite with a particular concentration of nanofiller. To facilitate disaggregation of ZnO nanoparticles, the 3 mL suspension of ZnO was sonicated (30 min at room temperature) before being added to the PSF solution. After stirring the ZnO suspensions in the PSF solution for 15 min, they were poured into the 5 cm^3^ reservoir of a commercial airbrush to subsequently spray them on the surface of flat aluminum foil. The amount of ZnO nanoparticles in the suspensions was selected to finally obtain nanocomposite materials with final concentrations (1%, 2%, 5% and 10%, % wt). A commercial gravity feed airbrush Elite E7116P Plus with the cup mounted at the top was used. The needle and the nozzle determine the amount of solution that can be sprayed through it. A 0.5-mm nozzle was used. Other conditions used to prepare the airbrushed films were 2 bars and 5 cm of air pressure and working distance, respectively. A flat, rectangular steel plate covered with aluminum foil was used as a substrate to prepare the films. All films prepared were stored in a desiccator.

In Figure 2, a scheme showing the commercial airbrush and the experimental set-up used, including the collector system, is shown. In Figure 2d, examples of representative thin films of PSF/ZnO prepared with the commercial airbrush are shown. As can be observed, highly transparent films were obtained, even for high loadings of nanoparticles. In this study, archaeological bones from the Magdalenian sequence of Coímbre cave in Besnes (Peñamellera Alta, Asturias, Spain) were used [36]. The bone samples were fragments found in the Upper Magdalenian in Coímbre (more precisely in Zone B) and dated to about 15,600 to 15,080 Cal BP. The Coímbre Cave bones have been exposed to a variety of taphonomical processes, including chemical alteration, a microbial attack that resulted in cracks and powdering. First macroscopic observations indicate that the bones were broken up and prone to breaking apart partially when handled. For this work, unidentified historical animal bone samples were chosen. The upper Magdalenian is dated from 15,000 to 13,200 cal BP [36]. To treat the surface of historical bones, the same airbrushing conditions were used. In Figure 3, a photograph of the historical bone samples after treatment with the airbrushed PSF/ZnO solutions is shown to illustrate the quality of the treated surfaces. In Table 1, the coding used to name the bone treated samples is included.

### 2.3. Characterization

#### 2.3.1. Structural and Morphological Characterization

A X-ray powder diffractometer, the Bruker ECO D8 Advance (Bruker, Karlsruhe, Germany), with a Cu, K_α1_ radiation was used in the range of angles 20° to 80° (2θ) using 2s per step and a step size of 0.02° to collect counts in a Bragg-Brentano configuration and coupled to a Lynxeye XE-T detector.

A TENEO field emission scanning electron microscope, FESEM (FEI), was used to investigate the morphology of nanoparticles and their distribution in the cross-section and surface of the polymer nanocomposites. The morphology was inspected by secondary (SE) and backscattered electrons (BSE). An acceleration voltage of 5.00 kV was applied, and a CBS detector was used. To avoid electrostatic accumulation, the samples were gold-coated by sputtering using a Leica EM ACE200 low vacuum coater (from Leica Microsystems S.L.U., Spain). Microanalysis by energy-dispersive X-ray spectroscopy, EDS, was also carried out to confirm the location of the different phases and their local concentrations.

To study the influence of the presence of ZnO nanoparticles in the polymer structure of the PSF polymer, the PSF/ZnO nanocomposites were characterized by Fourier-transform infrared (FTIR) spectroscopy using attenuated total reflectance, ATR. The spectra were recorded at room temperature in a Shimadzu Affinity 1 spectrometer equipped with a Golden Gate ATR accessory (diamond window), from 600 to 4000 cm^−1^ with a resolution of 4 cm^−1^ and averaging 32 scans (Shimadzu Europa GmbH). The software OMNIC ESP version 5.1 (Nicolet) was used for the numerical treatment of the spectra.

The thermal decomposition behavior of the PSF/ZnO nanocomposites was investigated by thermogravimetric analysis, TGA, using a TGA-SDTA 851 Mettler Toledo ((Mettler Toledo, Greifensee, Switzerland, distributors from Spain) by heating from 30 °C to 800 °C at a heating rate of 10 °C·min^−1^ under a nitrogen atmosphere with a gas flow of 20 mL·min^−1^. Moreover, from the TGA curves, the corresponding differential thermogravimetric analysis (DTGA) was carried out.

#### 2.3.2. Mechanical Characterization

The films were mechanically characterized using a Microtest DT/005/FR universal machine (Microtest S.A., Madrid, Spain) with a load cell of 50 N. In a uniaxial tensile setup, six specimens of each sample were tested at a loading rate of 1 mm·min^−1^. The specimens had a length of 4 cm, a width of 6 mm and a thickness of 35 µm on average. From the study of the tensile test curves, the mean values of several mechanical parameters, such as the elastic modulus, the tensile strength and the total deformation were obtained.

Local mechanical properties of the treated and untreated bones were carried out by Martens hardness measurements on the surfaces of the specimens. Martens hardness measurements were performed using Zwick Roel Z 2.5 hardness testing equipment (Zwick GmbH & Co., Ulm, Germany). The specimens were cut into 70 × 10 × 5 mm^3^ pieces, and ten hardness values from valid indentations were recorded for each material. According to the ISO 14577 standard for Marten hardness, the load and dwell time were set to 10 N and 3 s, respectively, and the speed of indentation until first contact with the specimen was set to 5 mm·min^−1^ [35]. 

## 3. Results and Discussion

### 3.1. Morphological Characterization

Knowing the distribution of nanofillers is crucial to understanding the possible unique features and properties of nanocomposite materials. In Figure 4, images of the surface and cross-section of the neat PSF and PSF/ZnO nanocomposites with different particle contents (0, 1, 2, 5, 10%, % wt) are shown. As can be seen, the neat PSF presents a smooth and uniform surface (Figure 4a). However, there is a kind of wavy pattern on the surface of the sample, as observed by visual inspection of the SEM images of the surface (see Figure 4a,c,e). Probably, that might be due to the solvent evaporation during the airbrushing procedure if there is not a completely continuous and uniform deposition of the upper layer of material. In relation to the cross-section micrograph of the neat PSF thin film (Figure 4b), it is possible to observe that there is plastic deformation that might have occurred during the preparation of the cross-sectioned sample.

SEM micrographs corresponding to the surface and the cross-section of the PSF + 1% ZnO material are shown in Figure 4c and d, respectively. Again, for this material, the surface is quite smooth and homogeneous since no particle agglomerates are observed. On the other hand, the cross-section morphology of this sample seems to be divided into two regions (purple highlighted). The first one, with a more porous microstructure, and the second one, with a more compact morphology. This last observation can have its origin in the mechanism of fracture. The smoother region may arise from the proper slippering of the razorblade used to prepare the cross-sectioned samples, while the second one can be due to a simple freeze-fracture. This change of fracture mechanism may happen because of the lack of continuity along the films since the process of preparation was in fact discontinuous. The deposition of material by airbrushing was carried out in two deposition steps, corresponding to the emptying of two reservoirs of the commercial airbrush of 5 mL each. The separation between the two regions with different microstructures was highlighted in purple (Figure 4).

In Figure 4e,f, SEM micrographs for the PSF + 2% ZnO sample can be seen. The surface morphology in this case is very similar to that observed for the sample with 1% ZnO NPs. When 2% ZnO is added to the PSF, there are some small aggregates on the surface of the sample, probably due to the ZnO NPs, Figure 4e. The image obtained for the cross-section of the PSF + 2% ZnO sample, Figure 4f, is also divided into regions corresponding to two layers with different mechanical responses to the freeze fracture when preparing the sample. In both images, bright spots can be observed, which can be attributed to the presence of ZnO nanoparticles and the sensitivity of the backscattered electrons to the atomic numbers of the elements. This assignment was confirmed by EDX microanalysis, which confirmed the presence of a much higher concentration of Zn in the bright regions on the SEM images. In Figure 5, the EDX microanalysis carried out on brighter regions of two different samples is shown, specifically for PSF + 10% ZnO (Figure 5, top) and PSF + 5% ZnO (bottom).

### 3.2. Structural Characterization

To understand possible specific interactions between PSF and ZnO nanoparticles, a detailed analysis of the FTIR spectra of the materials prepared was carried out. In Figure 6, the FTIR-ATR spectra of all samples under study are plotted. ZnO spectrum (Figure 6, top) shows a wide absorption band in the range 3000–3500 cm^−1^ assigned to O-H stretching of the hydroxyl groups with different surroundings. The absorption bands at 1027 cm^−1^ and within the range 800–600 cm^−1^ correspond to the presence of Zn–O bonds [37,38,39] thus making it possible to identify the presence of ZnO in the PSF/ZnO composites. The rest of the spectra show absorption bands centered at 1635 cm^−1^ and 1427 cm^−1^ and 1329 cm^−1^, usually associated with C=C and C-C stretching vibrations, respectively. The FTIR spectrum of pure PSF and its absorption bands correspond to the bottom spectrum of Figure 6. Pure PSF shows bands at 1020 cm^−1^ and 1103 cm^−1^ (aromatic C–H in-plane bending vibrations), 1244 cm^−1^ (C-O-C stretching vibration), 1151 cm^−1^ (O=S=O stretching vibrations), and 1292 cm^−1^ and 1321 cm^−1^ (S=O=S symmetric and asymmetric stretching vibrations) [40] and the peaks at 1490 cm^−1^–1585 cm^−1^ attributed to the aromatic stretching vibrations [41,42,43,44]. Upon the addition of ZnO nanoparticles, the spectrum of polysulfone did not display any changes in its characteristic absorption bands or shifts, thus suggesting that no covalent bonding or any other specific interactions exist between PSF and ZnO.

### 3.3. Thermal Characterization

The thermal stability of all materials considered was studied by thermogravimetric analysis (TGA). In Figure 7 and Table 2, the main results of TGA studies are gathered. In Figure 7, the plots of the mass loss as a function of temperature, thermogravimetric analysis (TGA) and their derivatives (differential thermogravimetric analysis, DTGA) are shown. A first step with a clear mass loss at around 160 °C is observed, which is related to the desorption of absorbed moisture or possible small amounts of solvent remaining in the polymer nanocomposite [45], which is somewhat unlikely if we consider that no traces of solvent are observed in the FTIR spectra (Figure 6). The second step, between 535–540 °C, where the main mass loss of the polymer occurs, corresponds to bulk polymer chain degradation [46,47]. In the DTGA curve, this mass loss corresponds to the peak temperature, (T_p_, °C) and it is the temperature at which the maximum degradation rate occurs, being one of the characteristic TGA parameters. After that, there is a final stage that is usually assigned to the slow decomposition of the PSF residue. This residual mass is common in the nitrogen atmosphere (pyrolysis), but it can be reduced if the experiment is conducted in air [46]. Additionally, the change in the final mass was calculated by comparing the final mass of each PSF/ZnO sample with that of pure PSF. In every case, the difference is higher than that corresponding to the amount of particles added, meaning that the presence of the particles lowers the degradation of the polymer. To evaluate the degradation rate in this final stage, the slope after the main mass loss was calculated and included with the rest of the thermogravimetric analysis parameters, labeling it as slope after pyrolysis (%·°C^−1^). In Table 2, the main data associated with TGA experiments are included. As can be observed numerically, the peak temperature, T_p_, increases slightly as particle content increases, from 535 °C for pure polysulfone to 539 °C for PSF filled with 10% ZnO nanoparticles. Although the increase in T_p_ with particle content is low, this result may indicate that the presence of ZnO nanoparticles slightly decreases thermal degradation of the neat polymer. Except for the amount of ZnO nanoparticles, there are not big changes in the TGA curve profiles, suggesting that the presence of the ZnO nanoparticles does not cause important changes in PSF structure, which is in accordance with FTIR results.

### 3.4. Mechanical Characterization

Six specimens of each composition (PSF/ZnO) were tensile tested to study how the addition of ZnO particles affected the mechanical behavior of PSF. In Figure 8, the stress-strain plots of the six specimens tested for all the materials under study are presented. For neat PSF films, regardless of the specimen, the maximum strain to failure was below 4% while the maximum tensile stress reached values of 14 and 16 MPa. In terms of mechanical behavior, when a relatively low amount of ZnO nanoparticles is added to the PSF (PSF + 1% ZnO films), a considerable increase in elastic modulus is observed. This result can be interpreted considering a good load transfer from the matrix to the hard and rigid particles due to a uniform dispersion of them, which facilitates ZnO-PSF interactions, leading to an effective reinforcement of the PSF. The elastic moduli of PSF + 2% ZnO, PSF + 5% ZnO, and PSF + 10% ZnO increased with particle content, although less pronouncedly than at low concentrations of nanoparticles. In these last cases, the incorporation of a stiffer material, like ZnO, is the main and simplest explanation for the increase in rigidity of the materials.

However, the presence of ZnO nanoparticles distributed throughout the polymer matrix might cause a hindering in the movement of the PSF molecular chains, reducing the interaction among the PSF macromolecules, thus making interactions among the chains more difficult [30,40,41]. On the other hand, the presence of aggregates and agglomerates of ZnO nanoparticles may also influence mechanical behavior and polymer chain interactions [48]. 

Upon the introduction of a small concentration of particles, 1% ZnO nanoparticles for instance (Figure 8), both the tensile strength and maximum strain to failure increase considerably compared to the pure polysulfone. This means that the addition of ZnO nanoparticles improves mechanical strength both in terms of tensile strength and toughness. A similar trend was observed when higher particle contents were considered. As particle content increases, the material becomes more resistant to stress and has higher plastic deformations (see Figure 8). In fact, the PSF + 5% ZnO and the PSF + 10% ZnO showed the highest strain to fraction and total area among all samples. However, to confirm these observations, a quantitative analysis of the stress vs. strain plots was conducted to obtain the corresponding mechanical parameters.

In Table 3, the information corresponding to different mechanical parameters calculated from the stress-strain curves is collected (tensile strength, elastic modulus, strain to failure, stress to failure and total area under the curve). As it is observed directly from the stress–strain plots, tensile strength generally increases as the concentration of ZnO nanoparticles increases. Tensile strength for pure polysulfone is ~9.3 MPa; it increases up to ~22 MPa for the sample with 1% ZnO and then increases steadily until ~28 MPa for the sample with 10% ZnO. Therefore, the introduction of ZnO nanoparticles considerably increases the mechanical strength of the polymer, in this case, polysulfone.

Regarding the elastic modulus, the data in Table 3 show that there is an increase in the elastic modulus from ~(541 ± 230) MPa for pure PSF to ~(822 ± 175) MPa for 1% ZnO NPs. After that, the modulus decreases, although not significantly, reaching a value of ~(803 ± 240) MPa for 2% ZnO NPs, then it follows ~(836 ± 221) MPa for 5% ZnO NPs and finally ~(1015 ± 361) MPa for the PSF + 10% ZnO nanocomposite sample. The results obtained for elastic modulus revealed that, as expected, the addition of ZnO results in an increase in the elastic modulus, meaning that specimens containing zinc oxide become stiffer, demonstrating that ZnO acts as an efficient reinforcing material.

Compared to pure PSF films, the nanocomposite films showed a considerable increase in elastic modulus that should be not only due to the incorporation of particles of high modulus but also to the quite uniform dispersion of them within the PSF polymer matrix, leading to good transfer of loads from the ZnO particles to the PSF matrix and vice versa, as it was described for other nanocomposite systems. When the increase in mechanical properties is not so evident, usually it is because there is important aggregation or even agglomeration of the nanoparticles [49]. The materials PSF + 10% ZnO and PSF + 5% ZnO showed the highest strain to failure, the highest area under the curve and the highest Young’s modulus (Table 3). These results can be explained by considering a stretching model similar to that observed in a former study of airbrushed polysulfone filled with hydroxyapatite nanoparticles [50].

To investigate the influence of ZnO particles on the elastic modulus of the materials, the upper bound (E_upper_) and lower bound (E_low_) were calculated using the rule of mixtures with the densities of the pure components (PSF ρ = 1.24 g·cm^−3^; ZnO ρ = 5.6 g·cm^−3^). The elastic modulus of PSF determined in this study (E = 541 MPa) [50] and the elastic modulus of ZnO was published (E = 111 GPa) [51]. The estimated values of the elastic modulus (E_upper_ and E_low_) were estimated.

In particle-reinforced composite materials, the elastic modulus can also be estimated using a modified expression of the rule of mixtures [50]. Values of the reinforcing efficiency parameter, K, were estimated for each PSF/ZnO system, these results are also included in Table 3. As the concentration of ZnO particles increases, K decreases, i.e., the reinforcing efficiency decreases. One possible explanation is that as particle content increases, aggregates and agglomerates are formed, resulting in a poorer reinforcing efficiency (see Figure 4). Therefore, even though a relatively uniform particle dispersion was observed for the samples with 5% ZnO NPs and with 10% ZnO NPs (Figure 4), the formation of some agglomerates appears. In addition, in the case of the sample with 10% ZnO NPs, the cross-section SEM micrograph revealed a highly porous section (Figure 4), which would also contribute to obtaining lower mechanical properties than those expected.

The strain and stress to failure finally exhibit a slightly different trend than tensile strength and modulus, as well as the total area under the curve. The general trend observed was that the strain to failure and stress to failure increased with increasing ZnO content. With the slightest inclusion of 1% ZnO NPs, the strain to failure and total area rise, probably because of the particle size and uniform dispersion of ZnO nanoparticles in the polymer matrix. Then, with increasing content of ZnO particles, the strain to failure increases, but more slightly. However, the total area in the 5 and 10% ZnO significantly increased. It can be concluded that ZnO, apart from acting as an effective reinforcing material, as shown by the stiffening of specimens when ZnO content increases, has an additional effect when aggregates and agglomerates are formed and, as was mentioned above, can be explained by the stretching model reported in reference [50].

### 3.5. Mechanical Characterization of the PSF/ZnO Surface Treatment: Martens Hardness Measurements

First, morphological observation of the cross-sections of the PSF/ZnO airbrushed bone samples was conducted. In Figure 9, the cross-section of the PSF/ZnO airbrushed bone treated samples is shown. The average thickness of the coatings is included. The thicknesses varied on the different samples, from approximately 7 µm to 37 µm, probably due to the heterogeneity of the bone surfaces. Here, it is possible to observe that the piece of bone treated with 10% ZnO is fractured, probably because the specimen selected for this study was highly damaged. In Figure 9, the micrograph on the right-hand side shows the X-ray microanalysis of a brighter region illustrating the presence of Zn, thus confirming the presence of ZnO nanoparticles.

Figure 10 illustrates representative force *vs.* indentation depth curves of tested ancient bones treated with PSF/ZnO. The results of Martens hardness (HM) are gathered in Table 4. The values of HM show an increase from 139.4 N·mm^−2^ for the untreated bones to 282.3 N·mm^−2^ for the treated bone with neat PSF. Then, by adding 1% ZnO to the PSF, the HM increases significantly, up to 850.1 N·mm^−2^. Apart from the introduction of 1% ZnO nanoparticles, the thickness of the coating should also be considered, as it is doubled from the sample 5P to the sample 5P-1Z. When the particle content increases to 2% ZnO, HM decreases to 625 N·mm^−2^ with respect to the coating with 1% ZnO. Although the particle content increases, in this sample (PSF + 2% ZnO), it is higher than that of the sample with 1% ZnO, the thickness of its coating is the smallest, with a value of approximately 7 µm, resulting in a decrease in the HM value. Then, as the concentration of ZnO nanoparticles increases, the HM decreases until a concentration of 10% ZnO, for which the HM is 173 N·mm^−2^. Probably this is due to the damaged surface of the samples.

Considering these results, it is observed that MH increases up to 1% ZnO nanoparticles, but after that, with a higher increase in the concentration of filler, the Martens hardness decreases remarkably. Apart from the effect of particle content, aggregation of nanofillers may affect the final mechanical behavior of nanocomposites rather than the interfacial adhesion between polymer and nanoparticles [52]. In fact, the production of nanocomposite materials may result in the aggregation/agglomeration of the nanoparticles because the probability of aggregation increases with higher particle loadings. The formation of these aggregates can result in defects and stress concentrations, thus reducing the mechanical strength of composite materials [53]. Therefore, by increasing the nanofiller content and decreasing the filler size, an increase in these effects might occur, leading to poorer surface mechanical properties.

## 4. Conclusions

The preparation by airbrushing and characterization of films based on polysulfone, PSF, filled with zinc oxide, ZnO, nanoparticles (NPs) were investigated. The novelty of this research relies on two viewpoints. First, the use of a commercial airbrush to prepare uniformly transparent PSF/ZnO thin films of approximately 40 µm. The second aspect relies on the use of this preparation method being considered as an alternative approach to in-situ depositing thin films of polymer nanocomposites for the consolidation and restoration of historical bones.

To accomplish these objectives, first, thin films of PSF/ZnO with different particle contents (0%,1%,2%, 5% and 10%, % wt) were prepared with a commercial airbrush. The mechanical characterization of the thin films showed that elastic modulus (E) increased with particle content, reaching an exceptionally high E for the sample filled with 1% ZnO nanoparticles. The presence of ZnO NPs did not modify the degradation temperature of the polysulfone, although a slight increase was observed with the content of ZnO. Therefore, it is possible to consider that the presence of the nanoparticles slightly hindered the thermal degradation of the polymer matrix.

Historical bone samples were airbrushed treated with different PSF/ZnO solutions, revealing that SBS is a feasible method to treat damaged samples. Special care should be taken when selecting the system composition, i.e., particle content. As particle content increases, mechanical properties in terms of stiffness and rigidity increase too. However, for high particle content, the formation of aggregates or agglomerates may lead to a decrease in mechanical properties. In terms of surface mechanical properties, for bone-treated samples, the best mechanical properties were obtained for the sample filled with 1% ZnO. These results allowed us to conclude that airbrushing polymer solutions is a good alternative for the consolidation of historical bones.

Apart from the use of airbrushing methodology for the restoration and conservation of historical bones, airbrushing methodology may have a broad field of uses and applications for the conservation and restoration of different artworks. For example, airbrushing can be used for the restoration of decorative objects, paintings, sculptures, furniture, etc., given its versatility as well as the possibility to be used for “in-situ” conservation and restoration applications in one specific location if necessary. Therefore, there is a broad spectrum of future applications not only in the field of conservation and restoration but also in other areas where localized “in situ” applications are demanded.

## Figures and Tables

**Figure 1 nanomaterials-13-00625-f001:**
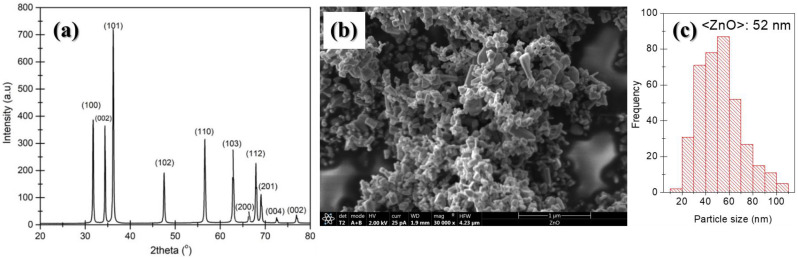
(**a**) XRD pattern of the as-received ZnO nanopowder and (**b**) SEM image of ZnO particles at 30,000×. (**c**) Particle size distribution obtained from SEM micrographs.

**Figure 2 nanomaterials-13-00625-f002:**
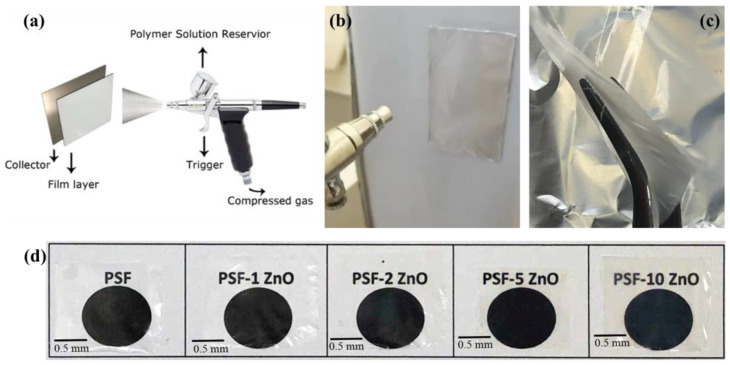
Sequence illustrating sample preparation: (**a**) airbrushing set-up; (**b**) photograph of the airbrushing on the surface of the aluminum foil; (**c**) extraction of the PSF/ZnO thin films and (**d**) photographs of the thin films of PSF/ZnO under study.

**Figure 3 nanomaterials-13-00625-f003:**
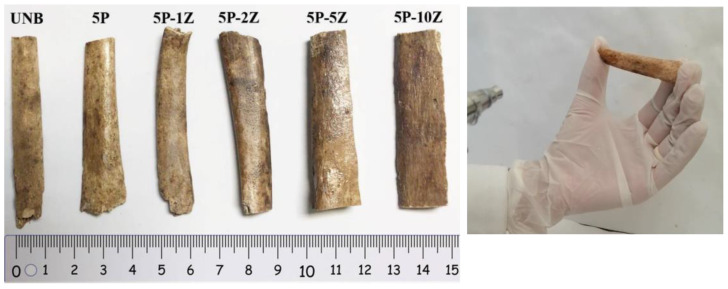
Airbrushed PSF/ZnO historical bone samples as a function of particle content (**Left**) and a photograph of the airbrushing process on the surface of historical bone (**Right**).

**Figure 4 nanomaterials-13-00625-f004:**
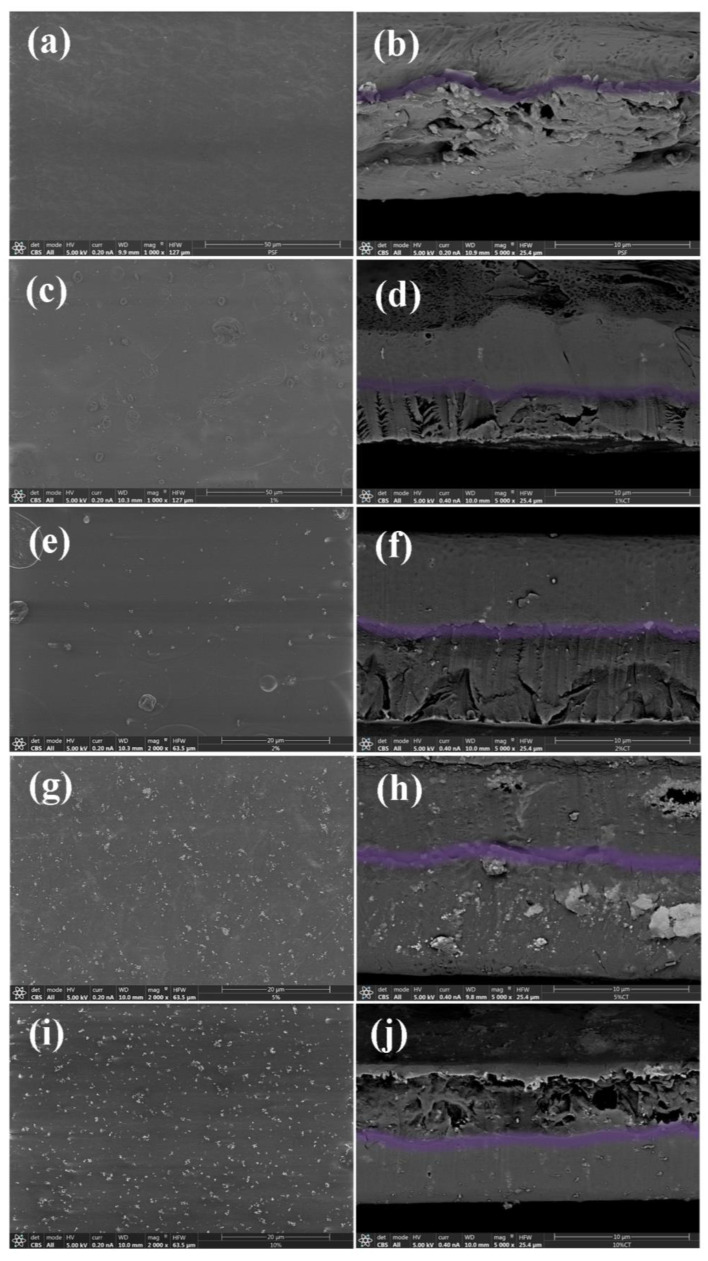
SEM images of the surface and a cross-section of the different PSF nanocomposites with different amounts of ZnO: (**a**,**b**) 0% ZnO; (**c**,**d**) 1% ZnO; (**e**,**f**) 2% ZnO; (**g**,**h**) 5% ZnO and (**i**,**j**) 10% ZnO (%, wt).

**Figure 5 nanomaterials-13-00625-f005:**
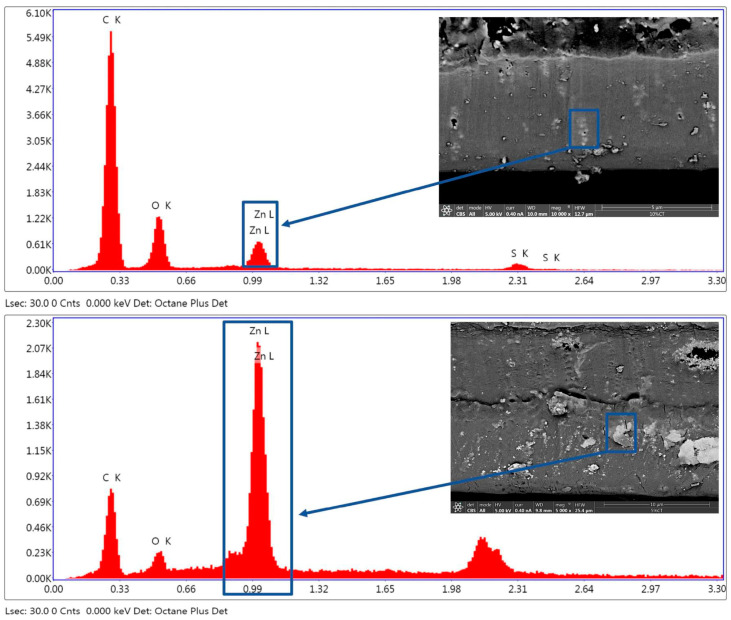
EDX microanalysis on the surface of two samples: PSF +10%ZnO (**top**) and PSF + 5% ZnO (**bottom**).

**Figure 6 nanomaterials-13-00625-f006:**
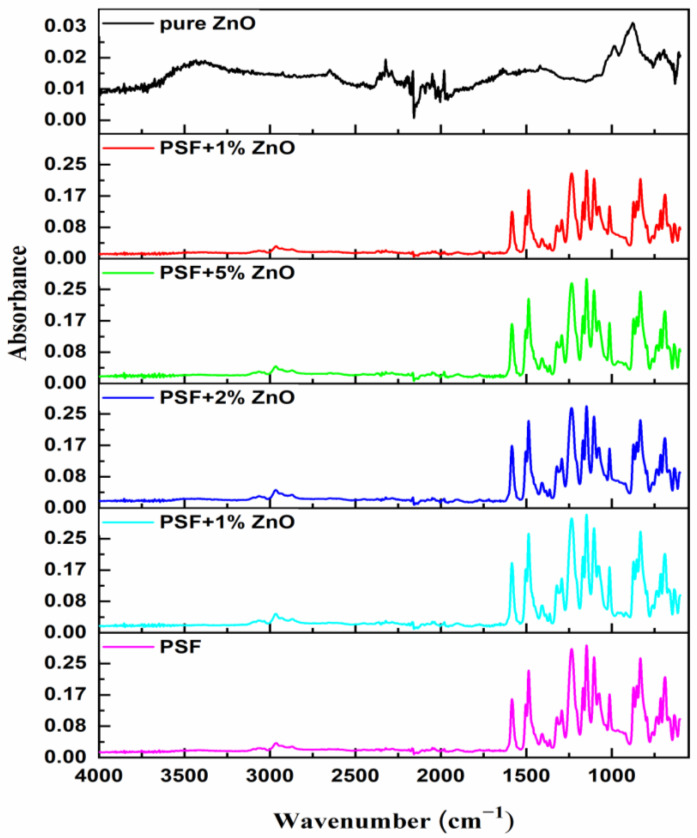
ATR-FTIR of PSF/ZnO film at the different concentrations of ZnO nanoparticles.

**Figure 7 nanomaterials-13-00625-f007:**
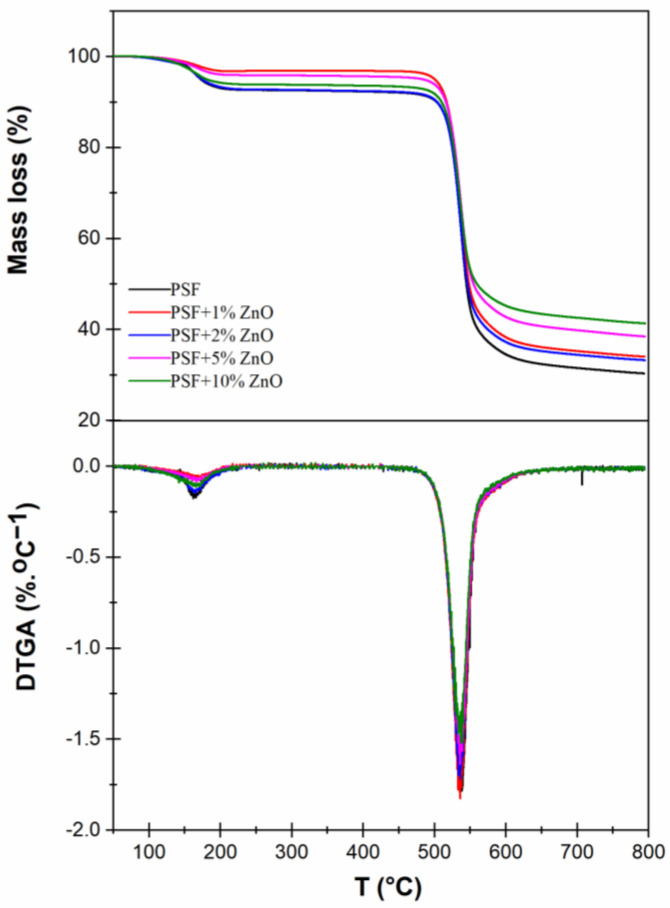
Thermogravimetric analysis, TGA and differential thermogravimetric analysis, DTGA of the PSF/ZnO nanocomposites.

**Figure 8 nanomaterials-13-00625-f008:**
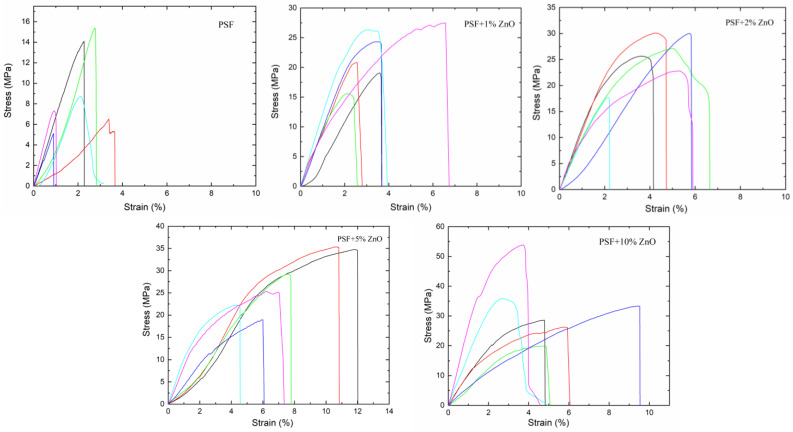
Stress-strain curves obtained for the PSF/ZnO thin film specimens uniaxially tested (The color lines correspond to each of the specimens tested).

**Figure 9 nanomaterials-13-00625-f009:**
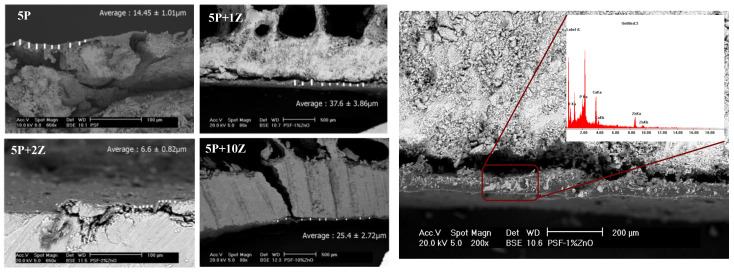
(**Left**): Cross-section SEM micrographs for the PSF/ZnO bone airbrushed samples. (**Right**): micrograph showing a X-ray microanalysis of a white region illustrating the presence of ZnO.

**Figure 10 nanomaterials-13-00625-f010:**
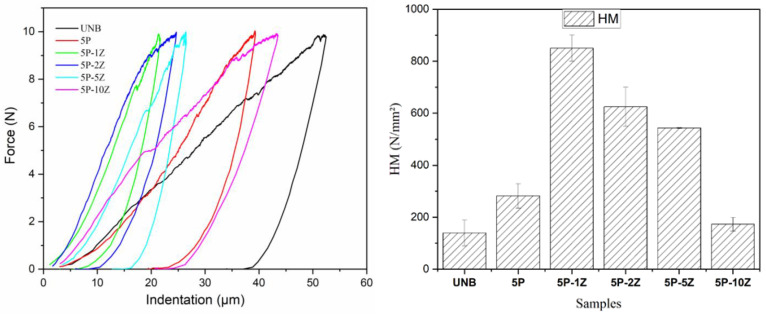
(**Left**): Representative force (N) vs. indentation depth (µm) curves obtained from Martens hardness experiments. (**Right**): average value of Martens hardness (HM) for the PSF/ZnO samples under study: UNB, 5P, 5P-1Z, 5P-2Z, 5P-5Z and 5P-10Z.

**Table 1 nanomaterials-13-00625-t001:** Sample code for airbrushed PSF/ZnO historical bone-treated samples.

Description of the Treatment ^1^	Sample Code
Untreated Bone	UNB
Treated sample with 5% PSF	5P
Treated sample with 5% PSF + 1% ZnO	5P-1Z
Treated sample with 5% PSF + 2% ZnO	5P-2Z
Treated sample with 5% PSF + 5% ZnO	5P-5Z
Treated sample with 5% PSF + 10% ZnO	5P-10Z

^1^ Note: 5% PSF (% g·mL^−1^); % ZnO (% wt, referred to total mass).

**Table 2 nanomaterials-13-00625-t002:** Parameters obtained from thermal characterization by thermogravimetric analysis (TGA).

Samples	Initial Mass	Final Mass			DTGA	
Sample Name	Mass (mg)	Mass (mg)	Mass (%)	Change in Final Mass (%) ^1^	Slope after Pyrolysis (%·°C^−1^)	T_peak_ (°C)
PSF	4.5354	1.3742	30.3	0	−1.78	535
PSF + 1%ZnO	6.1185	2.0375	33.3	3.0	−1.83	536
PSF + 2% ZnO	6.1494	2.0478	33.3	3.0	−1.72	536
PSF + 5% ZnO	9.1104	3.5066	38.49	8.2	−1.64	538
PSF + 10% ZnO	4.7485	1.9649	41.38	11.1	−1.55	539

^1^ Note: Change in the final mass (%) = (Mass of PSF + % ZnO) − (Mass of PSF).

**Table 3 nanomaterials-13-00625-t003:** Main representative mechanical parameters of the PSF/ZnO thin films from the uniaxial stress-strain tests.

Sample Name	σ (MPa)	E_exp_ (MPa)	E_upper_ (MPa)	E_low_ (MPa)	K	Strain to Failure (%)	Stress to Failure (MPa)	Total Area (10^6^ J·m^−3^)
PSF	9.52 ± 2.80	541 ± 229.7	546	541	0	2.31± 1.08	9.14 ± 4.13	0.1 ± 0.07
PSF + 1% ZnO	22.26 ± 4.36	822 ± 175.5	784	542.15	1.00	3.90 ± 1.43	21.91 ± 4.59	0.57 ± 0.35
PSF + 2% ZnO	25.45 ± 4.40	803 ± 237.8	1038	543.43	0.528	4.91 ± 1.52	23.03 ± 4.83	0.84 ± 0.33
PSF + 5% ZnO	26.13 ± 5.47	836 ± 220.9	1811	547.26	0.236	8.12 ± 2.69	25.87 ± 5.80	1.22 ± 0.57
PSF + 10% ZnO	27.98 ± 6.24	1016 ± 361	3192	554.24	0.124	6.43 ± 2.19	27.35 ± 5.59	1.28 ± 0.60

**Table 4 nanomaterials-13-00625-t004:** Results of the Martens hardness of the airbrushed PSF/ZnO ancient bone samples.

Sample	HM (N·mm^−2^)	h_max_ (μm)	HU_plast_ (N mm^−2^)	W_elast_ (Nmm)	W_elast_ (Nmm)
UNB	139.4 ± 50.2	52.0 ± 0.5	202.8 ± 29.4	0.05030 ± 0.0010	0.1860 ± 0.0460
5P	282.3 ± 47.4	36.7 ± 3.2	437.4 ± 89.8	0.047590 ± 0.0007	0.97524 ± 0.032
5P-1Z	850.1 ± 50.3	21.0 ± 0.5	1916.4 ± 172.2	0.04246 ± 0.0010	0.04613 ± 0.0042
5P-2Z	625.3 ± 75.7	25.0 ± 1.5	1248.6 ± 166.2	0.04512 ± 0.0063	0.06055 ± 0.0130
5P-5Z	543.5 ± 107.9	26.4 ± 2.3	940.3 ± 259.9	0.03593 ± 0.0012	0.08168 ± 0.0170
5P-10Z	173.3 ± 26.3	46.9 ± 3.7	321.3 ± 70.9	0.06990 ± 0.0026	0.01575 ± 0.0084

## Data Availability

The data of this study are available upon request.

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
