# Peer review of "Airbrushed PSF/ZnO Composite Coatings as a Novel Approach for the Consolidation of Historical Bones"

_nanomaterials, 2023, doi:10.3390/nano13040625_

Round 1

Reviewer 1 Report

Dear Authors,

The paper submitted for review by M. Moradienayat et al. titled Airbrushed PSF/ZnO coatings as a novel approach for the consolidation of historical bones concerns the practical application of nanocomposites for bone restoration treatments. In terms of application, the paper is quite useful, but the substantive approach to the topic is quite poor.

1.       Please describe how the nanometric particle sizes of ZnO were determined using scanning electron microscopy, usually DLS or TEM is used for this purpose, which allows to determine the actual particle values and not estimates (as in SEM/EDS)

2.       Please explain on what basis the Authors claim high dispersion of ZnO without showing a mapping with decomposition of the analytical element which could be zinc. Spot analysis is not evidence here - it can indicate both single particles and agglomerates

3.       On what basis do the Authors claim to have suspended nanoparticles in PSF/ZnO solution with any content; were viscosity tests done to show the stability of nanoparticle suspension in polymer or at least DLS of these suspensions?

4.       How long did the suspension sputtering last, how did the thickness of the PSF/ZnO layer change with time, was there a decrease in suspension flux observed over time or was the pressure on the gun increased?

5.       FTIR-ATR spectra for the studied nanocomposites are not clear, they do not show the changes the authors write about, the bands from ZnO overlap with those from the polymer. In addition, the ATR technique using the diamond attachment allows tracking changes only within one depth of penetration (4um), perhaps the Authors were not lucky enough to encounter a ZnO agglomerate. All the spectra visible in Figure 6 are the same, if you are writing about detailed analysis then please stretch the region,(800-600cm-1), where subtle changes are to be visible. Or change the crystal in the attachment to one that allows more accurate measurements. Currently, this study - as the spectra show - is unnecessary because it adds nothing to the work.

6.       If the authors see the presence of moisture on the basis of the DTA study, it means that you are unlikely to have a homogeneous dispersion there, but agglomerates, which can be seen in the SEM/EDX spot analysis (Fig 5) as well as in the mechanical characteristics (different nature of the curves within one material). A valuable study here would be to do DSC analysis rather than TGA, which would indicate the role of ZnO, for example, as a nucleant. The observed changes are the result of varying ZnO content in the PSF matrix vs. the role of the nucleant in relation to the PSF chain.

7.       He suggests looking carefully at the mechanical results and selecting repeated curves so as to reduce the error; currently, in many cases, it is at the level of 30% this means a completely random measurement and a layer that has nothing to do with the nanocomposite... The presentation of the results would be more valuable if the curves for different materials were put together on one graph and the shape analyzed - of course, in case the materials were homogeneous and not random as in your study. In addition, nanoindenters are used for nanocomposite layers rather than classical hardness testing methods dedicated to geological materials (Mertens hardness is derived from Mohs hardness) or even polymeric materials (Brinella method)

Author Response

Dear Reviewer, 

Thank you for your comments and suggestions. Please find attached a document with the answers provided. The manuscript was modified accordingly. 

Best regards.

Reviewer 2 Report

Dear authors,

this is an original piece of work, I do not find often a direct correlation between the polymeric restorational materials and bones as substrates. Yet, please see the comments below to improve your manuscript:

-title: proposition "Airbrushed PSF/ZnO composite coatings..."

-affilliation in English

-"free-standing thin films": what do you mean by that? I have not come across that term, avoid in Abstract, explain in text and rephrase if possible (line 92 too)

-l.13, 36: check spaces between numbers

-keywords: almost the same words in title. replace some with other important conslusions or methods used in paper

-l.31: prefer "ageing" spelling (check if apprears elsewhere in text)

-proper writing of polymers "poly(vinyl acetal), poly(vinyl butyral)"

-"antifungal, and UV filtering": never use a comma before "and" or "or" when a simple parathesis of similar thing in sentence. Check and correct in text where needed

-"In previous studies... can be maintained...": rephrase please, false tense/type for the verb maintain

-"larger interphases.... third component...": the interphase may be greater when the nanoparicles are added but in no case can be considered as a third component in the composite material! In that case, I would ask "what are the properties of the 3rd component?" That is not the case, the properties are derived from PSF and ZnO molecules only.

-"the photocatalyst activity": vocabular mistake "the photocatalytic activity"

-"model system": what do you mean? Model for new composites?

-Consider moving Fig. 1 and 2 in Results part, along with their comments. 

-l.125: 5% (w/v) or 5g/mL? I suppose the second is correct

-l. 138: proper writing "...and 10 %wt)"

-Table 1: I suppose "%wt" in all cases

-l. 182: "4000-600 cm-1", for IR always the great wavelegth first, it is left on the spectrum. Check for IR comments too (l. 260)

-"However, there is a kind of wavy pattern on the surface of the sample that might be due to the solvent evaporation during the airbrushing procedure": are you sure for that observation? By bare-eye or microscope? Explain the reason thoroughly

-Table 2: the columns "Mass (mg)" are unnecessary, remove. The column "initital mass %" too, all recordings in TGA start from 100%. Explain more in text the meaning and the detection of slope in pyrolysis plateau yet

-"thus making polymer chain-to-chain interactions more difficult": interactions among the chains, perhaps

-vs, ie: in italics, because of latin origin

-Regarding flexural experiments: it is generally accepted that the mechanical properties appear little reproducability. It is good you tried 5-6 repeats, but you see you got vast range in values. My opinion is to remove the too big or too low recordings and re-calculate with n=3 par example. 

-Table 3: remove the upper and the lower column of E values. Do you see the st.dev of 230, of 360 MPa? Is that acceptable?

-Table 4: Likewise, remove the upper and the lower values, the mean values give the character of the composites. The st.dev. of 50, 75, 107 N/mm2 are too big.

-Enhance Discussion with comments of the intercations between the polymer and the filler. Are there any H-bonds, Van der Waals forces? A scheme of the chemical structure of PSF would help.

-l.457: (...and 10 %wt)

-The source, origin, the nature of the bones? Carbonate, phospahte? 

Author Response

(The authors gave the same response as above.)

Round 2

Reviewer 2 Report

-Please rearrange the text, so less gaps are shown before/after the figures. 

-Fig. 8: what is the point of showing all recordings of specimens of a specific composite? In my opinion, choose the repetition that is close to the average behaviour of the composite, then make one chart that compares their behaviour.

-Table 2: initial and final mass (mg) is not needed, you may erase the columns. Slope pyrolysis with 2 decimals.

Author Response

First of all we would like to thank the reviewer the revision of the new version of the manuscript. We very much appreciate effort to improve the quality of the manuscript. Please find below our answers to this second revision.

Comment 1. Please rearrange the text, so less gaps are shown before/after the figures.

Answer: Thank you for the comment. The text was revised and the figures were adapted so that less gaps appear in the document.

Comment 2. Fig. 8: what is the point of showing all recordings of specimens of a specific composite? In my opinion, choose the repetition that is close to the average behaviour of the composite, then make one chart that compares their behaviour.

Answer: Thank you for the comment. Certainly, in Figure 8 were included all the curves of the mechanical test. However, taking into account that in mechanical testing experiments usually the dispersion is relatively high and moreover in composite materials. For this reason, we considered that, if the reviewer and the editors agree, it would be sensible to show the readership the kind of curves that were obtained. In such a way the readers can see

Comment 3. Table 2: initial and final mass (mg) is not needed, you may erase the columns. Slope pyrolysis with 2 decimals.

Answer: Thank you for the comment. In relation to Table 2, an additional column was added, labeled as “Change in the final mass (%)”. This mass was calculated by subtracting the final mass of pure PSF from the final mass of each sample. In this way, it is emphasized the effect that the nanoparticles exert on the thermal behavior of the composites.

The following text was added to the revised version of the manuscript: Additionally, the change in the final mass was calculated by comparing the final mass of each PSF/ZnO sample with that of pure PSF. In every case the difference is higher than that corresponding to the amount of particles added meaning that the presence of the particles lowers the degradation of the polymer.
